# A robot for overground physical human-robot interaction experiments

Sambad Regmi[1], Devin Burns[2], Yun Seong Song[1] *

**1** Department of Mechanical and Aerospace Engineering, Missouri University of Science and Technology, Rolla, MO, United States of America, **2** Department of Psychological Science, Missouri University of Science and Technology, Rolla, MO, United States of America

* songyun@mst.edu

## Abstract

Many anticipated physical human-robot interaction (pHRI) applications in the near future are overground tasks such as walking assistance. For investigating the biomechanics of human movement during pHRI, this work presents Ophrie, a novel interactive robot dedicated for physical interaction tasks with a human in overground settings. Unique design requirements for pHRI were considered in implementing the one-arm mobile robot, such as the low output impedance and the ability to apply small interaction forces. The robot can measure the human arm stiffness, an important physical quantity that can reveal human biomechanics during overground pHRI, while the human walks alongside the robot. This robot is anticipated to enable novel pHRI experiments and advance our understanding of intuitive and effective overground pHRI.

## Introduction

Present-day robots can perform specialized and sophisticated tasks around humans. Advancements in robotics technology have improved the robot's interaction with humans, in both social and physical interactions [1–4]. In particular, the development in the understanding of physical human-robot interaction (pHRI) has made a significant impact on how physically interactive robots can be employed in various aspects of human life [5, 6]. One of the many areas where these interactive machines may find their application is to provide assistance in overground walking. In contrast to those that don't have to interact physically with humans, these robots can always be questioned about their safety, intuitiveness, and effectiveness. For this, it is desired and expected that pHRI robots will have human-like interaction characteristics.

However, it is yet unclear what characteristics to instill in a robot to make it more human-like in overground pHRI. In physical human-human interaction (pHHI), humans do physical interaction solely through haptic-based information exchange even without any verbal communication [7–10]. Whether the contact is directly between humans, such as while dancing or assisting the elderly, or is through an object such as while moving a piece of furniture together, pHHI studies [7, 9–13] suggest that the motor intent is communicated in the form of coupled forces and movements. Other studies have identified human arm impedance modulation to be

**Data Availability Statement:** The dataset generated for this study can be found in the Harvard Dataverse [https://doi.org/10.7910/DVN/CC355J].

**Funding:** YS and DB Grant #1843892 National Science Foundation www.nsf.gov The funders had

no role in study design, data collection and analysis, decision to publish, or preparation of the manuscript. YS Grant #2046552 National Science Foundation www.nsf.gov The funders had no role in study design, data collection and analysis, decision to publish, or preparation of the manuscript.

**Competing interests:** The authors have declared that no competing interests exist.

a key phenomenon during pHHI tasks [9, 10]. Since impedance prescribes the dynamic relationship between forces and movements, these works suggest that the human arm impedance may be related to the motor communication during pHHI and pHRI.

Prior investigations of human arm impedance were focused on identifying its relationship with the motor control strategies of the upper arm in constrained seated pHRI tasks. At first, single joint impedance and muscle mechanical properties were studied using electromyography (EMG) signals [14, 15]. The multi-joint endpoint impedance of a human arm was first experimentally estimated from the recorded dynamic response of the arm after applying position perturbation during a maintained posture task [16]. Since then, the relationship between the modulation of endpoint arm impedance and how the central nervous system (CNS) executes the motor control strategies has been studied in relation to arm stability, learning novel dynamics, reaching tasks, and grasping force [11, 17–27]. In all these works, robots played a crucial role since they allowed both applying perturbations and recording forces and movements which provided a direct method to measure the human arm impedance.

In this view, a specialized robot capable of measuring and analyzing the dynamics of overground physical interaction is the key to understanding human motor communication during overground pHRI. This robot will be faced with unique conditions and challenges that are not present in seated pHRI studies mentioned above, such as increased kinematic freedom of human arm postures as compared to the fixed posture in [16]. Nonetheless, this robot should also possess the interactive-ness demonstrated by the seated pHRI robots in earlier studies, such as the low output impedance [5, 28] or the ability to provide precise perturbations [16, 17, 29].

To this end, this paper presents (a) the design and implementation of Ophrie—the robot for overground pHRI experiments (Fig 1), (b) an experimental demonstration of the interactive characteristics of the robot during overground pHRI, particularly in measuring the human arm stiffness, and (c) discussion on experimental settings with Ophrie that may affect future overground pHRI experiments. In addition, unique challenges of overground pHRI experiments are presented and discussed in the perspective of arm impedance measurement, which is a potentially impactful application of Ophrie in the near future.

## Materials and methods

### The "Ophrie" robot

The developed overground pHRI robot (Ophrie) consists of an interactive robotic arm and a wheel mobile base—that served the purpose of locomotion (Fig 1). It is expected to exchange interaction information with a human partner via hand, similar to the apparatuses used in seated pHRI experiments [6, 16, 28]. The unique requirements for an overground interaction for Ophrie were identified and discussed in detail in a prior work [30, 31]. These include the ability to apply and measure small interaction forces (<10 N) as well as inherently low endpoint impedance. The force requirement was achieved by selecting the appropriate actuators and sensors. The inherent impedance requirement was achieved by incorporating lightweight links and bearings and avoiding gear reductions. In order to verify this requirement as early as in the design phase, a computer simulation model of the robotic arm was developed, and a novel simulation method to estimate the endpoint impedance of a manipulator was proposed [31]. The feasibility of the designed arm in using it for the pHRI experiment was validated based on the estimated impedance of the robot arm.

A separate mobile robot capable of taking velocity commands was used for the robot base. The robot arm is lodged on top of the mobile base with a custom-made body frame in between for housing the electronics as shown in Fig 1. The body is height adjustable allowing human

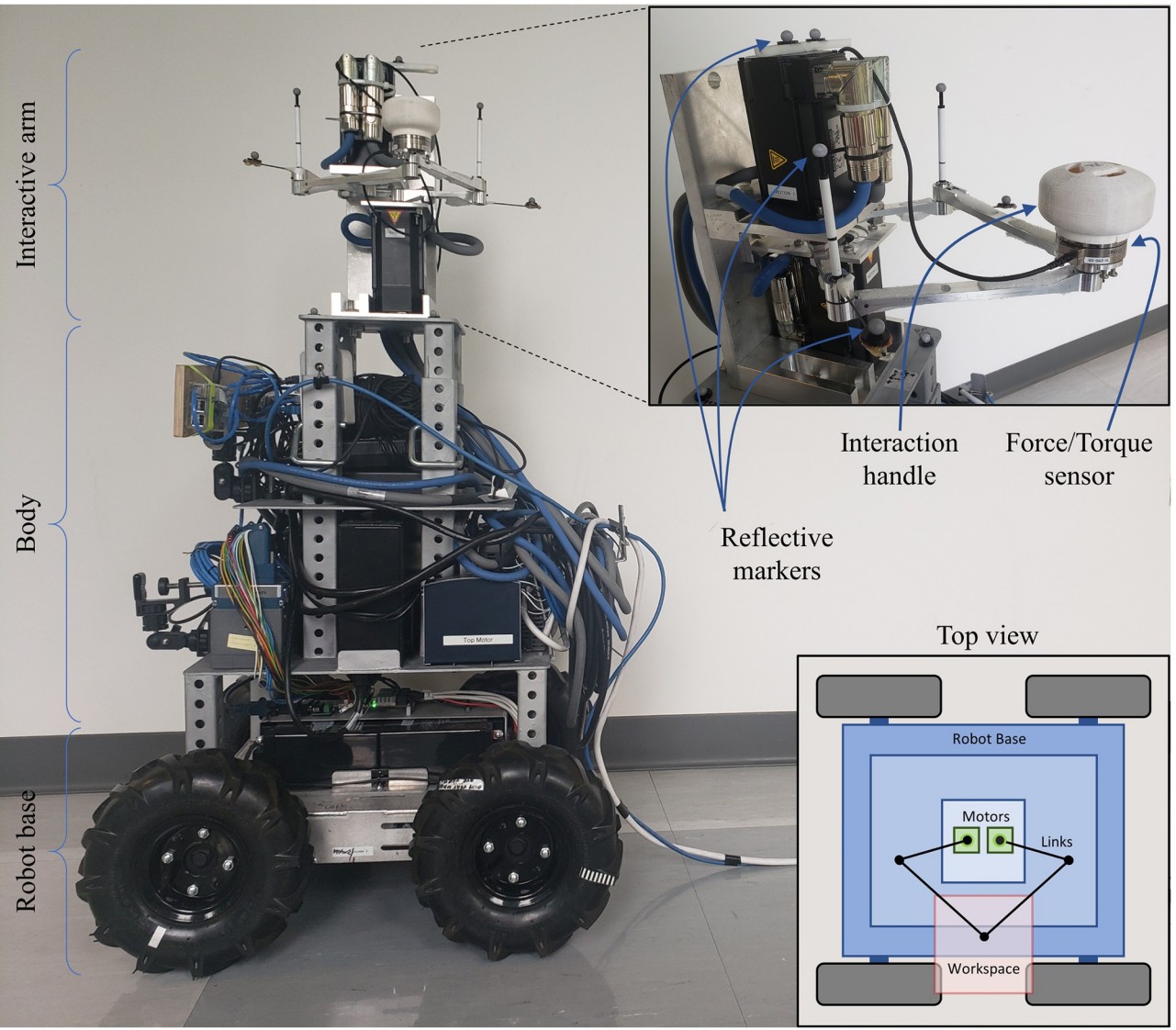

**Fig 1. Ophrie robot.** The picture of the robot with the interactive arm magnified in the inlet, and the schematic diagram of the top view shown in the bottom right.

subjects with varying heights to interact with the robot without discomfort that may arise from height differences. The robot is approximately 100 cm in height, which is smaller than other robots used in overground pHRI studies such as in [12].

**Ophrie's mechatronics system.** The interactive arm is a 2D closed-loop five-link mechanism formed by two equal-sized distal links (0.175 m), two equal-sized proximal links (0.110 m), and a ground link (0.040 m). This direct drive 2D mechanism is driven by two servo motors (AKM32E-ANCNAA00: Kollmorgen Corp., VA, USA) placed on either side of the ground link. Each servo motor requires a servo drive (AKD-P00606-NBEC-0000: Kollmorgen Corp., VA, USA) for power supply, encoder feedback, and control. These servo drives are controlled through a central real-time controller (cRIO 9045: NI, TX, USA). A force/torque sensor (mini45: ATI Industrial Automation, NC, USA) is lodged on top of the joint of two distal links at the end of the robotic manipulator. A custom-made interaction handle is placed on top of

the sensor as shown in Fig 1. The servo drives and force/torque controller are AC powered, whereas cRIO uses a 24V DC power supply through AC to DC converter. Two additional 24V power supply modules (PS-16: NI, TX, USA) are required to provide the logic power to the servo drives. All these mechatronics units are placed on the robot's body and are powered by an external AC source via wire.

The mobile base is a four-wheel differential drive robot (IG52-DB4 Heavy duty chain and sprocket kit: SuperDroidRobots, NC, USA). A 12V DC supply powers it through two recharge-able lithium-ion batteries seated on top of its frame. It is controlled using a micro-controller (RoboClaw 2x30A: Basicmicro Motion Control, CA, USA) which ultimately receives the command from the central controller.

The central controller is a central processing unit. It communicates with servo drives through a master-slave EtherCAT protocol and the base micro-controller through RS-232 serial communication protocol. It also accepts analog voltage data from the force sensor via force/torque controller (ATI Industrial Automation, NC, USA); a C-series module (9252 voltage input module: NI, TX, USA) is required to input voltage data to the controller. For monitoring the health of the force/torque controller, a separate digital input-output module (9401 module: NI, TX, USA) is required. Fig 2 shows the flowchart for Ophrie's mechatronic system.

LabVIEW programming language (LabVIEW 2018 with SoftMotion and Real-Time module: NI, TX, USA) is used to deploy the commands to the central controller from the remote computer. The designated computer and the real-time controller (on the robot) are connected through Ethernet maintaining the same subnet mask.

**Data acquisition.** To study the interaction dynamics during a pHRI experiment, it is essential to measure and record the interaction forces at the end effector as well as the subject's hand location kinematics. The forces at the end effector can be measured with respect to the force sensor axis and can later be transformed into the robot axis using an appropriate rotation matrix. The motor position can be recorded using a single-turn absolute Biss sine encoder (2048 LPR) embedded in the servo motors. The point of interaction, which is the end of the manipulator or the hand position, can be calculated from the motor positions by using the forward kinematics of the robotic arm. The data are recorded with a sampling rate of 1kHz.

When necessary, the arm length—as presented in section 'Observations from Ophrie-human interaction'—can be obtained from the Vicon motion capture system (Vicon Motion Systems Ltd., Oxford, UK). For this purpose, reflective markers can be placed in the hand and the shoulder. This data is then be processed afterward using Vicon Nexus and Vicon PorCalc Software. The sampling rate of the Vicon system is 200 Hz.

If the Vicon data were used for the analysis, the data from the robot was decimated to 200Hz before combining the two data. Then, the angular position provided by the motor

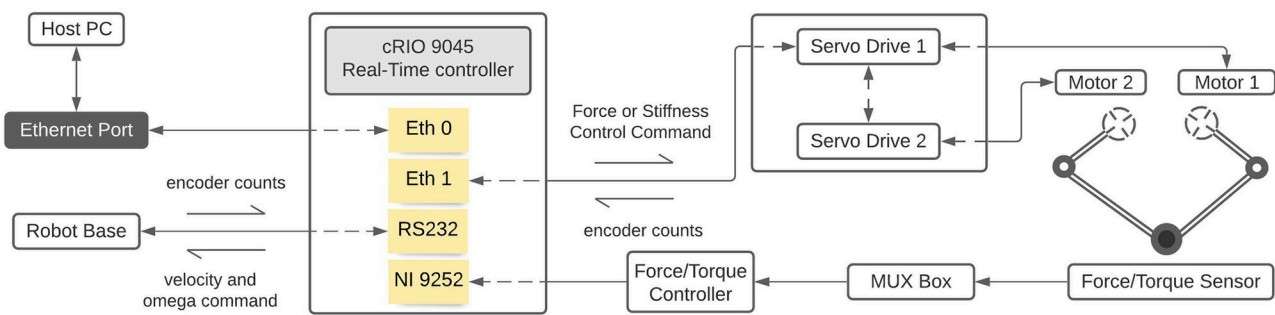

**Fig 2. The flowchart for Ophrie's mechatronics system.**

encoders was compared with the respective position obtained from the Vicon system, which is estimated with the help of 6 reflective markers attached to the robot arm (see Fig 1).

**Robot control.** The robot arm can be programmed to provide force or position modulation with PID control with 1 kHz. The robot can also provide force perturbations (<10 N) when needed. The position modulation can be used to provide a spring-like background force field towards the center of the workspace such that the handle position (and thus the human hand position) is kept away from the edges of the workspace. This background stiffness can be low (e.g., $50 \sim 100$ N/m) to allow smooth interaction with the user. Having this background stiffness is particularly important for an overground robot because, unlike in most seated pHRI tasks, it is very easy for the human user to move the handle outside the robot arm's workspace. Sufficient yet gentle background stiffness is thus crucial to help the endpoint of the robot arm to remain within the workspace during overground pHRI tasks.

The robot base is independently controlled and takes an input command of linear and angular velocity. The quadrature encoder reading from the motors of the robot's rear wheels is used for determining the instantaneous position, which later can be used for velocity control of the robot. The iteration rate for base control was 40 Hz.

## Validation experiment

Ophrie robot's ability to perform pHRI experiments and measure the dynamics of the interaction was validated with an experiment. Specifically, the pHRI experiment addressed Ophrie's ability to apply force perturbations to human hands and measure the resulting kinematics during overground pHRI. In addition, the experiment sought to investigate the effect of experimental conditions, such as human behavior, and robot settings in this specific pHRI experiment and suggest desired robot settings for future pHRI experiments.

A crucial and necessary feature of the robot is to allow arm stiffness estimation during overground pHRI tasks. To do this, Ophrie is designed to apply force perturbations and measure the hand displacement, similar to the methods used in [19, 32]. The procedure is described later in the section 'Experiment procedure' and the more detailed discussion can be found in the section 'Observations from Ophrie-human interaction'.

The robot settings may affect the interaction dynamics of the overground pHRI. Different levels of the perturbation force and the background stiffness of the robot (as explained in the section 'Experiment procedure') may affect the measurements. In addition, these settings may differently affect human behavior during pHRI depending on widely used experimental conditions, such as the availability of vision. To understand the possible interaction of robot settings with vision, this work included the availability of vision as the third condition.

**Subjects.** A total of 5 healthy young adults (3 male and 2 female, 27.4±1.817 years) without any self-reported neurological disorder took part in the study. Subjects signed a written consent form before participating in the experiment. The research protocol was approved by the Institutional Review Board of the University of Missouri System.

**Experiment procedure.** The experiment was designed to have twelve different cases defined by the various combinations of the following three variables: (a) visual condition (eyes open or eyes closed), (b) the background stiffness level of the robotic arm (50 N/m, 75 N/m, or 100 N/m), and (c) the level of force perturbation (3 N or 5 N). Trials were performed in four blocks, each randomly arranging the twelve cases mentioned above. The total number of trials per subject was 48. In this study, Ophrie acted as a leader, considering the complexities that have to be dealt with when the robot acts as a follower (see the section 'Observations from Ophrie-human interaction').

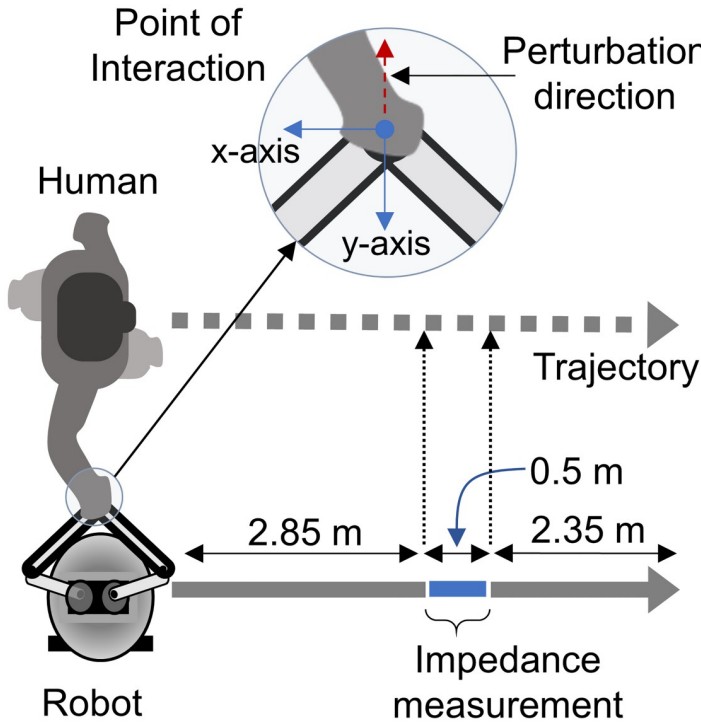

**Fig 3. Schematic of the overground pHRI experiment.**

After necessary preparation for motion capture, subjects were detailed about what was expected from them during the experiment. The robot's height was then adjusted to align the interaction handle to the subject's elbow level. Subjects were informed that the robot would stop if the interaction handle goes out of the workspace—a $0.15\times0.15\text{m}^2$ region inside which the subjects were asked to keep the hand while interacting with the robotic arm. The presence of the background stiffness helped with this. The level of background stiffness differed for different trials based on experiment design. At the beginning of each trial, the subjects were asked to stand upright next to Ophrie by holding the interaction handle (see Fig 3). Subjects were informed whether they had to keep their eyes closed or open for the upcoming trial and then were asked to shake the handle three times in the y-axis (see Fig 3) for data synchronization purposes. The process was followed by activating the background stiffness controller; it positioned the robotic arm to the center of the workspace. Subjects were asked to relax the arm and let Ophrie position its interaction handle to the center of the workspace during this process. Lastly, they were asked to start walking alongside Ophrie, as shown in Fig 3. The robot was programmed to maintain a constant linear velocity of 0.5 m/s throughout the 5.7 m straight-line trajectories. At the midway of the trajectory, the robot applied a force perturbation (rectangular pulse with 1 second duration) in the axis perpendicular to the robot's movement direction and towards the interacting subject (- y in Fig 3). The stiffness control loop was disabled and replaced by the force controller during the perturbation. For each trial, the amount of force perturbation was different. The magnitude was determined by adding (i) the interaction forces due to the background stiffness control at the instant when the perturbation was to be applied and (ii) the level of perturbation commanded (i.e., 3 N or 5 N). Subjects were asked to follow the robot with a relaxed arm throughout the entire trial except during the

perturbation period in which they were asked to keep their hand within the workspace of the robot arm. Each trial took approximately 20 seconds.

For each trial, the manipulator's endpoint position (which is identical to the human hand position), the manipulator's endpoint velocity (which is identical to the human hand velocity), and the interaction forces between the robot and the human hand were obtained. The endpoint position and forces were measured directly from the motor encoders and the force sensor, respectively. The position data was filtered with a zero-phase digital filter in both forward and reverse directions with a low-pass cutoff frequency of 40 Hz using Matlab (The Math-Works Inc., MA, USA) to reject high frequency electrical noise and to keep low frequency human movement data. The velocity was obtained from the position data using numerical differentiation.

**Estimation of the arm stiffness.** From the obtained forces and kinematics, the interaction dynamics can be estimated. For initial validations, this work used one of the simplest and widely used models, which involves passive, linear, and time-invariant second-order dynamics, such that:

$$f - f_0 = m \ (\ddot{x} - \ddot{x}_0) + b \ (\dot{x} - \dot{x}_0) + k \ (x - x_0) \tag{1}$$

where $f$ is the interaction force measured after the perturbation onset and $x$, $\dot{x}$, and $\ddot{x}$ are the position, velocity and acceleration of the interaction point after the perturbation, respectively. $m$, $b$, and $k$ are the endpoint impedance parameters namely inertia, damping and stiffness, respectively [33–35]. Here, $m$ incorporates the inertia of both the human arm and the interaction handle, whose mass is ($\sim 0.05$ kg). $x_0$, $\dot{x}_0$, and $\ddot{x}_0$ are the mean hand position, velocity, and acceleration of the interaction point just before the perturbation, respectively. Since the length of the data for analysis is short (on average, 335.636 ms—see Results), the effect of the musculoskeletal system as well as the effect of short-term neuromechanical response are captured as the lumped impedance parameters in this linear model [19]. The data of 0.1 second window before the perturbation onset was used for averaging and are used as the operating point from which all deviations are measured. For simplicity, only the kinematics in the direction of the force perturbation are considered, such that the parameters $m$, $b$, and $k$ as well as the kinematics $x$, $\dot{x}$, $\ddot{x}$, $x_0$, $\dot{x}_0$, and $\ddot{x}_0$ are scalars.

The measured kinematics and forces were fitted on the Eq 1 using Matlab fitlm function to estimate $m$, $b$, and $k$. The intercept was constrained to zero to match the linear model in fitlm with Eq 1. Otherwise, default settings were used such that modelspec was set to linear and the observation weights were set to one. The stiffness value ($k$) was the main outcome of interest, while the inertia ($m$) and damping ($b$) values were used to verify that the regression result was acceptable (e.g., both $m$ and $b$ are positive and realistic). This regression method is similar to the ARX model regression used in [35]. Further details and justifications can be found in the discussion section 'Assumptions and challenges', and in the supplementary material. Lastly, the equilibrium point for each trial was obtained by applying the estimated $m$, $b$, and $k$ values into Eq 2, which represent the interaction dynamics just before the perturbation:

$$f_0 = m \ \ddot{x}_0 + b \ \dot{x}_0 + k \ (x_0 - x_{eq}) \tag{2}$$

where $x_{eq}$ is the position of the equilibrium point of the hand.

**Statistical analysis.** The experiment was conducted in four blocks with a randomized complete block (RCB) design. Each trial from the experiment produced one scalar stiffness value along the axis of the perturbation (y-direction, see Fig 3), which was always perpendicular to the robot movement. Out of 240 trials across all subjects, five trials that resulted in a negative stiffness and one other trial whose data was lost due to exporting error were discarded.

From the remaining data, outliers—defined as the values three times the inerquartile range past the 25th and the 75th quartiles—were screened out using JMP statistical software (SAS Institute Inc., NC, USA). There were a total of 3 outliers that were excluded from further analysis. The effects of the robot settings (background stiffness level and the magnitude of the perturbation force) and the effect of the experimental condition (eyes open/closed) were tested on the remaining 231 trials using ANOVA (JMP statistical software). The hypothesis was that the robot settings do not affect the overground pHRI dynamics, but the availability of vision affects the human response and alters the overground pHRI dynamics. A significance level of $p = 0.05$ was used.

## Results

All subjects successfully completed all 48 trials without fatigue or safety concerns. They quickly became accustomed to the protocol. There were no noticeable obstructions to the experiment due to the hesitance or reluctance of the subjects.

As explained in the section 'Experiment procedure', the background stiffness is present during the entire trial except when the perturbation is applied. The average interaction force of all the subjects and all trials due to background stiffness just before the perturbation was observed to be 2.063±1.108 N at 0.746±1.529 rad counterclockwise from the interacting subject's mediolateral direction (i.e.,—y in Fig 3). That is, on average, the subjects were pushing the robot away from their trunk at approximately 2 N of force. This force is similar to the interaction force between two humans in a similar task to this study (overground walking while holding a sensor together) [36] and lower than the interaction forces between two humans while walking side-by-side on treadmills [37].

The participants' hand positions before the onset of the perturbation were near the origin of the workspace as seen in Figs 4, 5B and 5D. The hand velocity was nearly zero at the onset of the perturbation (Fig 5A and 5C). A typical pattern was observed in the hand velocity time series in almost all trials of all subjects. Shortly after the force perturbation onset, the hand velocity along the y-direction decreased as the hand was pushed away from the robot until it reached a negative peak. Then the velocity increased towards the robot until it passed zero and reached a positive peak at around $250 \sim 450$ ms after the onset of the perturbation. We did not observe any noticeable pattern across trials after the second peak, but this pattern of negative-to-positive peaks was present in all trials in all subjects. For example, for a trial of subject 5, the interaction force, handle displacement, and the velocity after the perturbation onset is shown in Fig 6. For this specific trial, the second peak of the velocity was at 274 ms (Fig 6C). The average time of the second peak of all the trials was observed to be 335.646±81.676 ms for all subjects.

This pattern resembles the velocity response of a passive second-order system given a step input. Thus, we hypothesize that the period between the onset of the perturbation to the second peak of the velocity contained the passive dynamics and can be used to analyze the interaction dynamics (Eq 1). The interaction dynamics were estimated using linear regression as discussed in the section 'Estimation of the arm stiffness'. The estimated stiffness was found to be 235.255± 113.581 N/m, 199.893± 92.442 N/m, 174.958± 97.043 N/m, 149.107± 67.358 N/m, and 123.195±63.871 N/m, respectively for subjects 1 to 5. Since the robot arm does not have inherent stiffness elements, this stiffness may be interpreted as the stiffness of the arm of the human user. Indeed, the values were comparable with the arm stiffness reported in the previous seated experiments (200–300 N/m in [22, 34], 100–150 N/m in [16, 19]). Although the experimental conditions differ, these experiments include conditions that are comparable to those in this study, such as having relaxed arms, externally applied perturbation forces, or

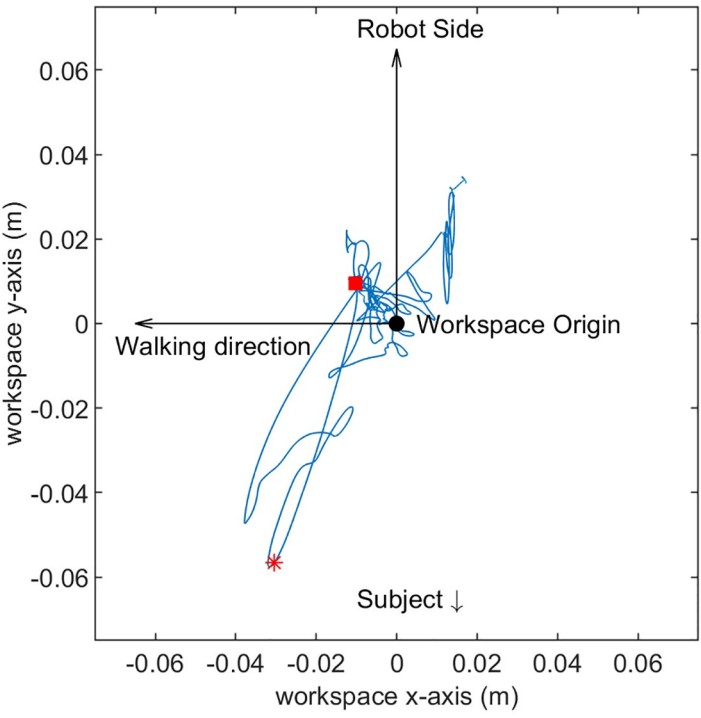

**Fig 4. The trajectory of the interaction handle/hand in 2D for a representative trial performed by subject 5.** Red square represents the instant when force perturbation initiated, and the red star represents the handle/hand position after 200ms of the perturbation.

varying arm postures. Nonetheless, several factors can affect the arm stiffness measurements between prior work and this work. For example, the difference in arm posture between seated experiments (the hand is in front of the subjects at shoulder height) and in our walking experiment (the hand is to the right side and on the waist level) may affect arm stiffness measurements [16]. Other factors such as having fixed versus free shoulder locations or the varying levels of challenge in these experiments may affect the arm stiffness measurements. The average of inertia estimated from all 48 trials of all the subjects was found to be 0.473±0.249 kg. The average damping was observed to be 3.270±4.741 Ns/m. The low positive values of inertia and damping were as expected since the combined inertia and damping of the robot arm and human arm would not be very high. The inertia, damping, and stiffness estimated for all 48 trials performed by subject 5 are plotted in Fig 7A–7C, respectively. In addition, the equilibrium point of the participant's arm per trial was also estimated. On average, the equilibrium point was 0.031±0.031 m away from the center of the manipulator workspace towards the robot. The mean and standard deviation of the $R^2$ values of the estimated dynamics of all 48 trials for each subject were 0.521±0.083 (subject 1), 0.489±0.072 (subject 2), 0.559±0.124 (subject 3), 0.549 ±0.050 (subject 4), and 0.567±0.092 (subject 5). These values are higher than the $R^2$ values typically reported in biomechanics literature, such as [38].

A two-way ANOVA between subjects and trial groups (earlier—first 24 trials, and later— last 24 trials) was performed to check for a possible learning effect between earlier trials and later ones. After observing no significant interaction effect between subjects and trial groups (p>0.219), we found no significant difference between the trial groups (p>0.869), indicating no apparent evidence of learning or adaptation as the trial number increased. The stiffness

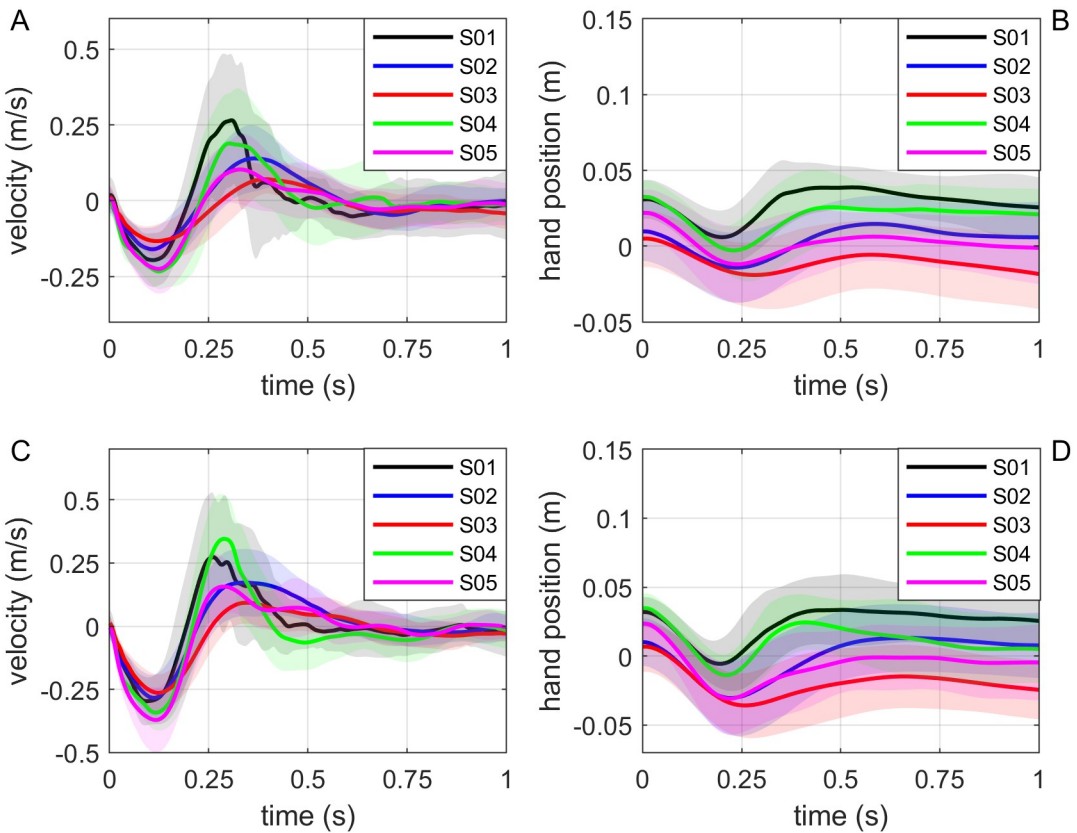

**Fig 5. Mean and standard deviation of hand velocity, and hand position with respect to workspace center for all five subjects during the perturbation period (1 sec).** (A) and (B) represent trials with 3N perturbation, and (C) and (D) represent trials with 5N perturbation.

parameter for trials belonging to the earlier and latter groups was 176.924.591 N/m and 174.360 N/m, with a standard deviation of 103.407 N/m and 88.430 N/m, respectively.

The overground pHRI experiment presented in this work includes three fixed effects (visual condition—2 levels, level of force perturbation—2 levels, and level of stiffness in the robot arm—3 levels) and one random effect (subject). A linear mixed effects model was used which included all possible interactions between the fixed effects but no random slopes representing interactions between the subject and the three fixed effects (which would have required a much larger sample size to have appropriate power). This model showed that the total variance due to the random effect (subject) was 18.466% (Table 1a). The model also showed that the main effects of visual condition ($p = 0.172$), the level of perturbation force ($p = 0.867$), and the level of background stiffness ($p = 0.288$) were all not significant, implying that neither the robot settings nor the availability of vision affected the human arm stiffness measurement (Table 1b). The interactions between the fixed effects were also not significant ($p > 0.093$). In particular, the availability of vision did not significantly interact with the robot settings. The effect of the robot settings on arm stiffness in all subjects can be seen in Fig 8.

## Discussion

### Observations from Ophrie-human interaction

Overall, the robot base's movement and the robot arm's dynamics were accepted well by the subjects. The forward velocity of the robot (0.5 m/s) appeared adequate while the subjects

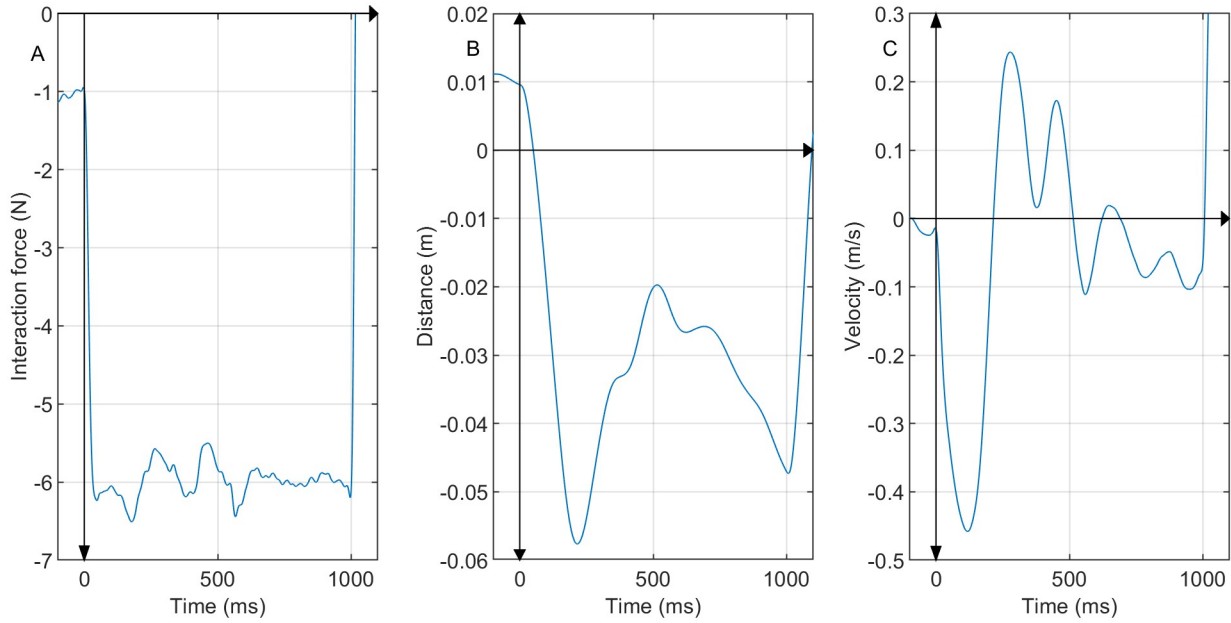

**Fig 6. Measurements during force perturbation of a representative trial of subject 5.** (A) The interaction force between the robot arm and the human subject, (B) the position of the hand/handle with respect to the origin of the workspace in y-direction, (C) the velocity of the handle in y-direction.

comfortably followed Ophrie. The background stiffness of the robot arm (50 to 100 N/m) was able to keep the human hand position near the origin of the robot arm's workspace most of the time. However, in some trials, the perturbation force pushed the hand to the outside of the workspace, in which case the trials were discarded and repeated. Although it rarely occurred, this happened more with low background stiffness (50 N/m) and high perturbation force (5 N) when the human hand position at the onset of the perturbation was in the $-y$ direction. Since the robot settings did not significantly affect the interaction dynamics (Table 1b), it is recommended to avoid the lowest background stiffness (50 N/m) and use the lower perturbation force (3 N) settings for future experiments with Ophrie.

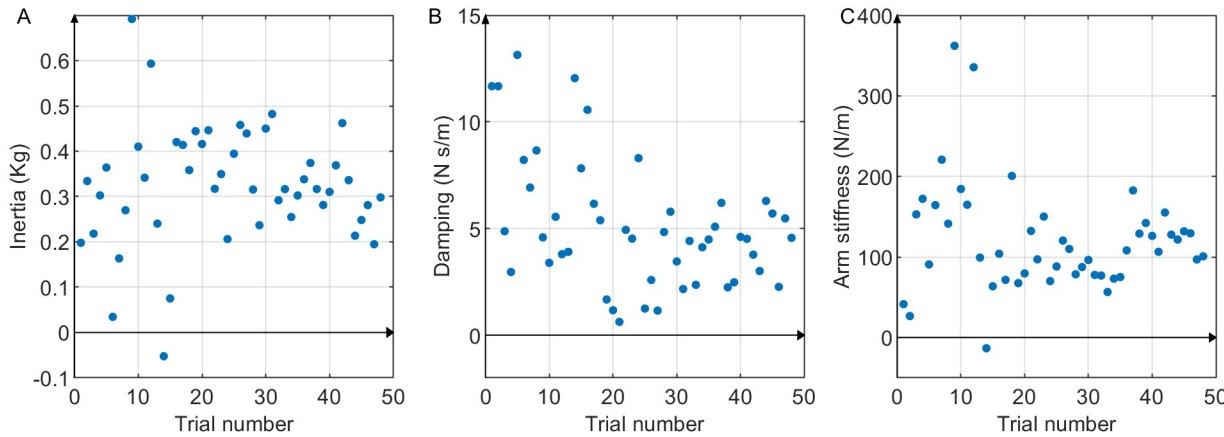

**Fig 7. The interaction dynamics of the subject 5 in y-direction in all trials.** (A) inertia, (B) damping, and (C) stiffness components identified through Eq 1.

**Table 1. The results from the linear mixed model.** The model analyzed the effect of the random effect of subject as well as the fixed effects for the level of perturbation (LP), the level of background stiffness (BS), and the vision (VC), on human arm impedance measurement.

(a) The effect of Subject on variance

| Random Effect | Var Ratio | Var Component | Std Error | 95% Lower | 95% Upper | Wald p-value | Pct of Total |
|---|---|---|---|---|---|---|---|
| Subject | 0.226 | 1763.438 | 1368.258 | -918.299 | 4445.175 | 0.1975 | 18.466 |
| Residual | | 7786.105 | 752.724 | 6498.329 | 9500.798 | | 81.534 |
| Total | | 9549.543 | 1553.549 | 7109.888 | 13508.998 | | 100.000 |

(b) The main effects and the interactions of fixed effects

| Source | Nparm | DF | DFDen | F Ratio | Prob>F |
|---|---|---|---|---|---|
| Visual Condition (VC) | 1 | 1 | 214 | 1.876 | 0.172 |
| Level of Perturbation (LP) | 1 | 1 | 214 | 0.028 | 0.867 |
| Background Stiffness (BS) | 2 | 2 | 214 | 1.251 | 0.288 |
| VC*LP | 1 | 1 | 214 | 0.016 | 0.899 |
| VC*BS | 2 | 2 | 214 | 2.406 | 0.093 |
| LP*BS | 2 | 2 | 214 | 0.185 | 0.831 |
| VC*LP*BS | 2 | 2 | 214 | 1.124 | 0.327 |

The equilibrium point of the human arm was estimated to be 0.031±0.031 m from the workspace center towards the robot. Fig 5B and 5D show that the average hand position right before the perturbation for all the subjects was towards the estimated equilibrium point. That is, the preferred equilibrium position of the hand was not at the center of the workspace where the interaction force would have been zero. This would not be a result of the human standing too close to the robot, because they were free to choose however close or far they could be with the robot. Instead, they all preferred to maintain a small pushing force between them and the robot. Since the equilibrium point location is decided by the central nervous system [39], it could be argued that it is the natural preference of humans to maintain a gentle push with Ophrie.

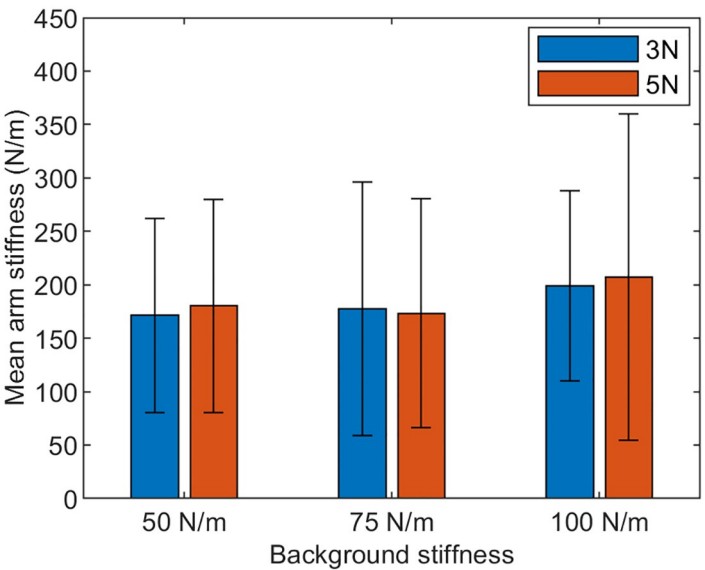

**Fig 8. Average stiffness and standard deviation among all subjects for different robot settings.**

The unique challenge in arm stiffness measurement during overground interaction tasks (compared to the traditional seated interaction tasks) comes from the lack of stationary 'ground'. During seated pHRI experiments, the upper body of the subjects is stationary and assumed to be the reference from which the hand position can be obtained. The change in arm length due to perturbation is thus equal to the difference in the hand position at before and after the perturbation. On the other hand, during overground pHRI tasks, it is possible for the shoulder to also move during the force perturbation at hand. However, while following Ophrie by holding its handle as it moves forward with no turning, with the help of the arm movement recording from Vicon camera system, we observed that the interacting human partner's shoulder move in a sinusoidal pattern in the mediolateral direction (due to the mediolateral sway during walking). As a result, the unperturbed arm length also followed the sinusoidal pattern since the shoulder position predominantly determined the arm length. When a force perturbation is applied at hand, this sinusoidal pattern may shift in response to the force. However, it was found that the shoulder did not deviate from its original sinusoidal pattern in the presence of the force perturbation. That is, the shoulder position seemed unaffected by the perturbation at hand. Hence, in our experiment, the change in arm length is equal to the change in hand position, which can be obtained from the encoders on the motors even when the shoulder position is not fixed with respect to the robot. In such a case, the shoulder and hand position recording (such as from Vicon camera system) is not required. We also did not observe any traces of shoulder movement (such as an obvious cyclic pattern) in the hand movement or interaction force data collected by the robot.

Since this work is the first to provide arm stiffness measurements in overground pHRI experiments, little was known regarding the possibility of the robot settings interacting with common biomechanical experimental conditions, such as the availability of vision. For example, our recent work [38] suggested that lowering arm stiffness would be beneficial for sensing interaction dynamics. If so, lowering arm stiffness would be beneficial for scenarios with increased uncertainty such as having no vision as well as low background stiffness. Our prediction was that, with no vision, the arm stiffness would be lower during low background stiffness trials compared to the arm stiffness during high background stiffness trials. But with vision, there would be little to no difference in arm stiffness between different background stiffness levels since the subjects can see the ongoing interaction. Contrary to our prediction, the vision condition did not interact with any of the robot parameters and especially with the background stiffness. This implies that at least within the provided robot parameter set, the availability of vision does not significantly affect arm stiffness measurements. For designing future overground pHRI experiments with Ophrie, one may use any combination of robot settings used in this study with or without vision. The same is true for different robot setting. Hence, any combinations of background stiffness (50, 75, or 100 N/m), perturbation level (3 or 5 N), or vision condition (eyes open or closed) could be used to measure the arm stiffness while designing an experimental protocol for the overground pHRI experiments in future. Nonetheless, as mentioned earlier, it was preferred by the study team to avoid the lowest background stiffness (50 N/m) as well as the higher level of perturbation (5 N) to minimize the possible occurrence of the hand being pushed out of the workspace of the robot arm.

Many pHHI and pHRI work discuss the effect of roles (leader or follower) in dyadic tasks [40]. In this work, Ophrie may not be a suitable follower since the wheeled base of Ophrie is not ideal for feedback control for adapting to quick changes in the commands (e.g. slow control loop). In addition, the wheel controllers have a predefined velocity profile along which the velocity increases or decreases. Nonetheless, this is acceptable for when the robot is leading and the human partner is following. While a hierarchical controller that takes advantage of the

fast and accurate robot arm control may allow Ophrie to take a follower role in the future, the current work focused on assigning the leader role to the robot. This also helped reduce variability in the experimental trials that may arise from variable robot trajectories. Having Ophrie as the leader also simulates a potential pHRI task of assisting people to walk.

## Estimating the dynamics during overground pHRI experiments

**Perturbation method.** A perturbation is provided for estimating the dynamics of physical interactions while the response is collected and interpreted. Two types of perturbations exist—force or position perturbations. Position perturbation has widely been used for human arm stiffness estimation since it was first introduced in mid 80's [16, 17, 20–27, 29, 34]. Position perturbation is useful when the expected movement trajectory of the hand is known, such as when the arm is at rest or when the unperturbed trajectory is provided [21]. However, in overground experiments similar to one presented in this work, predicting the unperturbed trajectory of the human arm/hand is nearly impossible due to the variability of the human movement while walking alongside the robot (as seen in Fig 4). Hence position perturbation may not be applicable.

Given this unique situation in overground pHRI, force perturbation is the only alternative for estimating the interaction dynamics; the recorded resultant displacement and the known applied force can be used for system identification assuming the arm dynamics to be linear second-order model. Force perturbation may be applied in the form of continuous perturbations with rich frequency components. [18, 41] used pseudo-random binary sequence (PRBS) force inputs continuously for the entire period of the trial to estimate the joint mechanical properties in limb motion tasks. These studies used multiple repeated trials to obtain the ensemble mean trajectory from which deviations are calculated for estimating the impedance parameters. Alternatively, time-frequency analysis was performed using force pulse to measure arm stiffness—which eliminated the requirement of multiple trials. For example, [42] used the estimates of body segment parameters that require a rich understanding of the inertial characteristics of the arm. In contrast, another [43] did not require such parameters. It used a parametric and a non-parametric estimator to estimate the continuous-time linear time-varying system of the arm dynamics. These methods require continuous mechanical perturbations in mostly stationary body segments. However, for the proposed pHRI experiment in this work, the arm configuration can neither be consistent within a trial nor between the trials. Hence the continuous force perturbation may not be applicable.

Because of these constraints, this work tested the feasibility of applying a single force perturbation instead of continuous force pulses and then using system identification similar to the methods used in previous studies by [19, 32]. This would not intervene with the continuous overground physical interaction process throughout the entire trial as well as eliminate the need for baseline unperturbed trajectory. To this end, in this work, we propose a method to estimate the stiffness within one trial, as presented in the section 'Estimation of the arm stiffness'.

**Assumptions and challenges.** The biggest challenge of the above method is to ensure that, at the time of the perturbation, the state of the system is similar across trials. For example, if the perturbation was applied at the middle of the trial, the arm configuration better be identical at that instant across all trials. However, it is nearly impossible to achieve this during overground pHRI tasks because the arm cannot be constrained due to the nature of the task. As a result, in this experiment, the perturbation may have been applied to the hand at varying arm configurations. This may have resulted in the stiffness parameter's relatively large variance

along with inter-subject variabilities such as the muscle cross-sectional area and co-contraction. Nonetheless, acknowledging the possible sources of deviations, larger sample sizes (or the number of trials) may help discern subtle changes in the parameters between experimental conditions. Indeed, as shown in Table 1b, statistical significance can be tested and discerned despite the potential variability.

A substantial assumption of the proposed method of estimating the dynamics of overground pHRI is that the dynamics is passive second-order as represented in Eq 1. While simple, this model, which was previously used in [41], served as a starting point in estimating the pHRI dynamics during overground tasks for the first time. Other studies, such as [33–35], have also used a similar mathematical model to represent the human arm dynamics. Consistent with the assumed model, the velocity response showed the characteristics of passive second-order dynamics between 0 ms and 250–400 ms. This allowed the estimation of the inertia, damping, and stiffness parameters of Eq 1. The estimated stiffness values were reasonable, with only five occurrences of below-zero stiffness out of 239 trials. These few aberrant values may have occurred due to the inherent variability of the overground pHRI experiment. The estimated damping and inertia values were also reasonable, although the variability seemed higher than for stiffness (Fig 7). This was somewhat expected since the first- and second-derivatives of the hand position are inevitably noisier than the position data. Hence, in this work, we used these parameters as indicators of the performance of the linear regression and the resulting stiffness estimate. The estimated inertia values were mostly positive and similar to the expected inertia of the lower arm ($0.3 \sim 0.5$ Kg). Also, the average damping was low and mostly positive, as expected. It was also comparable to the damping reported in [35]. Thus we concluded that the proposed method for estimating the dynamics is informative despite the simplicity.

## Estimated stiffness

As mentioned earlier, the estimated stiffness from Eq 1 may be interpreted as the stiffness of the human arm due to the inherently low mechanical impedance of the robot arm and the absence of a passive elastic element in the robot. When the force perturbation is applied in order to measure the arm stiffness, the CNS may modulate the arm stiffness in response to it. Hence the arm stiffness before and after CNS intervention may differ. In this view, our stiffness estimation method in the section 'Estimation of the arm stiffness' helps to include the pre-CNS intervention stiffness of the arm since the method uses the arm displacement data before typical neural delay (approx. $50 \sim 200$ ms) even though the dynamics hasn't settled down to a steady-state.

In the seated pHRI experiments, it was reported that the human CNS tends to increase the arm stiffness in the direction of unstable dynamics or disturbance [22, 24, 25]. For example, in [22], the arm stiffness was higher in the direction of instability (up to 750 N/m) compared to when there was no disturbance (as low as 160 N/m). In this view, if the additional degree of freedom of the arm in overground pHRI (allowed by the unconstrained upper body of the human subject) was interpreted as an additional factor aiding instability, it may show up as increased stiffness. However, the measured arm stiffness in y-direction was similar to the lower values in [22]. This implies that the CNS may not view the increased freedom of arm movement in overground pHRI as a drawback. Instead, it may maintain the low arm stiffness for the arm to be more sensitive to the small changes in the interaction forces [38]. However, because the stiffness estimation in this work warrants further development, the implications of such low arm stiffness values during overground pHRI remain a topic for future research.

### Prospective benefits

When physically interacting robots were first introduced for seated pHRI experiments, they quickly became invaluable tools not only for research but also for rehabilitation and physical therapies [5, 44]. We expect Ophrie to provide a similar impact and benefit to overground pHRI research. Key research questions may be addressed such as motor communication during overground physical interaction or to develop intuitive and effective walking assistance in the future. The demonstration of Ophrie's ability to perform an overground pHRI experiment and to estimate the interaction dynamics in this work is an important first step.

## Supporting information

**S1 Appendix. Interaction dynamics of the combined human arm-robot manipulator system.**
(PDF)

## Author Contributions

**Conceptualization:** Sambad Regmi, Devin Burns, Yun Seong Song.

**Data curation:** Sambad Regmi.

**Formal analysis:** Sambad Regmi, Devin Burns.

**Funding acquisition:** Yun Seong Song.

**Investigation:** Sambad Regmi, Yun Seong Song.

**Methodology:** Sambad Regmi, Devin Burns, Yun Seong Song.

**Project administration:** Yun Seong Song.

**Resources:** Yun Seong Song.

**Software:** Sambad Regmi.

**Supervision:** Devin Burns, Yun Seong Song.

**Validation:** Sambad Regmi, Yun Seong Song.

**Visualization:** Sambad Regmi.

**Writing – original draft:** Sambad Regmi.

**Writing – review & editing:** Sambad Regmi, Devin Burns, Yun Seong Song.

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
