## [Decision Letter · Decision Letter 0]

18 Aug 2022

PONE-D-22-07060A robot for overground physical human-robot interaction experiments

PLOS ONE

Dear Dr. Song,

Thank you for submitting your manuscript to PLOS ONE. After careful consideration, we feel that it has merit but does not fully meet PLOS ONE’s publication criteria as it currently stands. Therefore, we invite you to submit a revised version of the manuscript that addresses the points raised during the review process.

The manuscript has been evaluated by two reviewers, and their comments are available below.

The reviewers have raised a number of  concerns regarding the methodology, reporting and statistical analysis of this study. 

Could you please revise the manuscript to carefully address the concerns raised?

We look forward to receiving your revised manuscript.

Kind regards,

Johannes Stortz

Staff Editor

PLOS ONE

Journal Requirements:

Reviewers' comments:

Reviewer's Responses to Questions

**Comments to the Author**

1. Is the manuscript technically sound, and do the data support the conclusions?

Reviewer #1: Yes

Reviewer #2: Partly

2. Has the statistical analysis been performed appropriately and rigorously? 

Reviewer #1: Yes

Reviewer #2: No

3. Have the authors made all data underlying the findings in their manuscript fully available?

Reviewer #1: Yes

Reviewer #2: No

4. Is the manuscript presented in an intelligible fashion and written in standard English?

Reviewer #1: Yes

Reviewer #2: Yes

5. Review Comments to the Author

Reviewer #1: The choice of designing a robot for interaction experiments in motion is original, but some more details on the possible uses of this configuration would be useful.

The position of the hand is estimated based on the measurements of the motors, as well as the speed and acceleration. With regard to these last two quantities, has a simple numerical derivative been carried out or have filtering techniques been applied?

Is it possible to detect any traces of the sinusoidal movement of the shoulder in the experiments?

By force perturbation is meant a rectangular impulse?

The typical delay of the physiological response, even if mentioned by the authors, was not considered in the impedance model, a few more comments about this would help.

Reviewer #2: The main contribution of this paper is the presentation of the design of a new mobile platform with a robotic arm attached to it, used for the measurement of the impedance of the human arm during walking. To this reviewer’s knowledge, this is the first mobile platform used for this purpose, and the implementation is significant and of interest. This paper also provides the results of a preliminary experiment, where the same measurement of arm impedance is repeated in a number of conditions, i.e., under different visual conditions (eyes closed, eyes opened), and for different levels of perturbation and background stiffness.

This reviewer has a couple of major concerns:

1) Because this is a newly developed device, it would be good to validate the measurements of arm impedance vs. previous studies that quantified impedance in non-walking conditions. I think that an experiment in absence of walking (perhaps in a similar sitting position) would be useful to confirm the validity of the measurement of arm impedance obtained with this device. As the authors suggest, a lot of factors including the dynamics of the perturbing arm may affect the value of the measurements, but no validation is pursued for these target metrics.

2) I understand that the main focus of the paper is the presentation of the device and showing that it can measure arm impedance, and that the measurements are valid (see point above). However, there are a couple of concerns for the validation experiment pursued. I understand that the goal of the study is to understand how perturbation parameters affect the outcome, and they end up showing that they don’t (at least in a significant way), but what is the rationale for exploring the effect of visual condition? What was the hypothesis motivating the analysis, the rationale for forming such hypothesis, and the prediction?

Also I have a concern about the statistical analysis used by the authors. You use here a full-factorial ANOVA, but subject should be considered a random effect rather than a fixed effect. A linear mixed model is more appropriate for this purpose given that the same participants are tested in multiple conditions. The study may not be powered to detect high-level interactions between the random effects and all combinations of the fixed effects, and these high-level terms may be removed, especially in absence of a strong rationale for how these high level terms would be interpreted.

6. PLOS authors have the option to publish the peer review history of their article (what does this mean?). If published, this will include your full peer review and any attached files.

Reviewer #1: No

Reviewer #2: No

---

## [Author Response · Author response to Decision Letter 0]

6 Oct 2022

Thank you for the constructive reviews. The responses to the individual comments can be found in the uploaded document, 'response to reviewers.pdf'.

---

## [Editor Report · Decision Letter 1]

18 Oct 2022

A robot for overground physical human-robot interaction experiments

PONE-D-22-07060R1

Dear Dr. Song,

We’re pleased to inform you that your manuscript has been judged scientifically suitable for publication and will be formally accepted for publication once it meets all outstanding technical requirements.

Kind regards,

Gianni Ferretti

Guest Editor

PLOS ONE

Additional Editor Comments (optional):

The time scale in fig. 6 should be slightly increased, so as to make directly visible the trend of the rectangular force impulse